# TEXT EMBEDDINGS FOR RETRIEVAL FROM A LARGE KNOWLEDGE BASE

## ABSTRACT

Text embedding representing natural language documents in a semantic vector space can be used for document retrieval using nearest neighbor lookup. In order to study the feasibility of neural models specialized for retrieval in a semantically meaningful way, we suggest the use of the Stanford Question Answering Dataset (SQuAD) in an open-domain question answering context, where the first task is to find paragraphs useful for answering a given question. First, we compare the quality of various text-embedding methods on the performance of retrieval and give an extensive empirical comparison on the performance of various non-augmented base embedding with, and without IDF weighting. Our main results are that by training deep residual neural models specifically for retrieval purposes can yield significant gains when it is used to augment existing embeddings. We also establish that deeper models are superior to this task. The best base baseline embeddings augmented by our learned neural approach improves the top-1 recall of the system by $14\%$ in terms of the question side, and by $8\%$ in terms of the paragraph side.

## 1 INTRODUCTION

The goal of open domain question answering is to answer questions posed in natural language using an unstructured collection of natural language documents such as Wikipedia. Given the recent successes of increasingly sophisticated, neural attention based question answering models such as Yu et al. (2018), it is natural to break the task of answering a question into two subtasks as suggested in Chen et al. (2017):

- Retrieval: retrieval of the paragraph most likely to contain all the information to answer the question correctly.
- Extraction: Utilizing one of the above question-answering models to extract the answer to the question from the retrieved paragraphs.

In our case, we use the collection of all SQuAD paragraphs as our knowledge base and try to answer the questions without knowing which paragraphs they correspond to. We do not benchmark the quality of the extraction phase, but study the quality of text retrieval methods, and the feasibility of learning specialized neural models text retrieval purposes. Due to the complexity of natural languages, a good text embedding that represents the natural language documents in a semantic vector space is critical for the performance of the retrieval model. We constrain our attention to approaches that use a nearest neighbor look-up over a database of embeddings using some embedding method.

We describe the implementation details utilizing the pre-trained embeddings to create semantically more meaningful text embedding improved by utilizing an extra knowledge base, and hence improving the quality of a target NLP retrieval task. Our solution is based on refining existing text embeddings trained on huge text corpora in an unsupervised manner. Given those embeddings, we learn residual neural models to improve their retrieval performance. For this purpose, we utilize the triplet learning methodology with hard negative mining.

The first question studied here, is whether having advanced, and refined embedding may provide higher recall for retrieval in the external knowledge base. Therefore we benchmark most popular available embedding models, and compare their performance on the retrieval task. First, we start

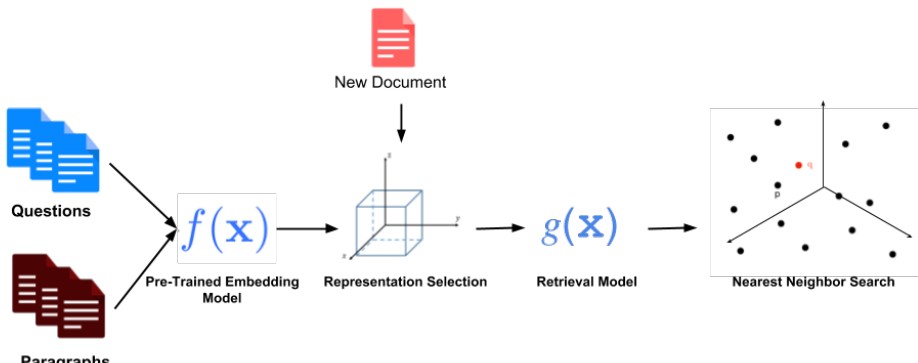

Figure 1: The workflow of the proposed approach for retrieval.

with a review of recent advances in text embeddings in section 2. In section 3 we describe our proposed approach. The core idea is to augment precomputed text embedding models with an extra deep residual model for retrieval purposes. We present an empirical study of the proposed method in comparison with our baselines which utilize the most popular word embedding models. We report the results in section 4. Finally, we conclude the paper with some future work in section 5.

## 2 RELATED WORK

There are various type of applications of word embedding in the literature that was well covered by Perone et al. (2018). The influential Word2Vec by Mikolov et al. (2013) is one of the first popular approaches of word embedding based on neural networks that was built on top of guiding work by Bengio et al. (2003) on the neural language model for distributed word representations. This type of implementation is able to conserve semantic relationships between words and their context or in other terms surrounding neighboring words. Two different approaches are proposed in [28] to compute word representations. One of the approaches is called Skip-gram that predicts surrounding neighboring words given a target word and the other approach is called Continuous Bag-of-Words that predicts target word using a bag-of-words context. As a following study, Global Vectors (GloVe) by Pennington et al. (2014), points to reduce some limitations of Word2Vec by focusing on the global context instead of surrounding neighboring words for learning the representations. The global context is calculated by utilizing the word co-occurrences in a corpus. During this calculation, a count-based approach is functioned unlike the prediction-based method in Word2Vec. On the other hand, FastText, by Bojanowski et al. (2017), is also announced recently. It has the same principles like others that focus on extracting word embedding from large corpus. FastText is very similar to Word2Vec except handling each word as a formation of character n-grams. That formation accelerate FastText to learn representation more efficiently.

Nonetheless, the important question still exists on extracting high-quality and more meaningful representations: How to seize the semantic, syntactic and the different meanings in different context. This is also the point where our journey is getting started. Embedding from Language Models (ELMo),by Peters et al. (2018), was newly proposed in order to tackle that question. ELMo extracts representations from a bi-directional Long Short Term Memory (LSTM),by Hochreiter & Schmidhuber (1997), that is trained with a language model (LM) objective on a very large text dataset. ELMo representations are a function of the internal layers of the bi-directional Language Model (biLM) that outputs good and diverse representations about the words/token (a CNN over characters). ELMo is also incorporating character n-grams like in FastText but there are some constitutional differences between ELMo and its predecessors. The reason why we focused on ELMo is that it can generate more refined and detailed representations of the words due to utilizing internal representations of the LSTM network. As we mentioned before, ELMo is built on top of a LM and each word/token representation is a function of the entire input which makes it different than the other methods and also eliminates the others' restrictions where each word/token is a mean of multiple contexts.

There are some other embedding methods which are not used as a baseline in the field; therefore, we did not get into them in detail. Skip-Thought Vectors, by Kiros et al. (2015), uses encoder-decoder RNNs that predicts the nearby sentence of a given sentence. The other one is InferSent, by Conneau et al. (2017), that utilizes a supervised training for the sentence embeddings, unlike as in Skip-Thought. They used BiLSTM encoders to generate embeddings. p-mean, by Rücklé et al. (2018), is built to tackle the unfairness of the Skip-Through that is taking the mean of the word/token embeddings with different dimensions. There are some other more methods such as Le & Mikolov (2014)'s Doc2Vec/Paragraph2Vec, Hill et al. (2016)'s fastSent, and Pagliardini et al. (2018)'s Sent2Vec.

Furthermore, we also include the newly-announced Universal Sentence Encoder by Cer et al. (2018) in our comparisons. Universal Sentence Encoder is built on top of two different models. Initially, the Transformer-based attentive encoder model, by Vaswani et al. (2017), was implemented to get high accuracy and the other model takes the benefit of deep averaging network (DAN), by Iyyer et al. (2015). As a result, averages of words/token embeddings and bi-grams can be provided as input to a DAN where the sentence embeddings are getting computed. According to the authors, their sentence level embeddings exceeds the performance of transfer learning using word/token embeddings alone.

Last, but not least, distance metric learning is designed to amend the representation of the data in a way that retains the related points close to each other while separating different points on the vector space as stated by Lowe (1995), Cao et al. (2013), and Xing et al. (2002). Instead of utilizing a standard distance metric learning, a non-linear embedding of the data using deep networks has shown an important improvements by learning representations using triplet loss by Hadsell et al. (2006), Chopra et al. (2005), contrastive loss by Weinberger & Saul (2009), Chechik et al. (2010), angular loss by Wang et al. (2017), and n-pair loss by Sohn (2016) for some of influential studies presented by Taigman et al. (2014), Sun et al. (2014), Schroff et al. (2015), and Wang et al. (2014).

After providing brief review of the latest trends in the field, we describe the details of our approach and experimental results in the following sections.

# 3 PROPOSED APPROACH

## 3.1 EMBEDDING MODEL

After examining the latest advanced work in word embedding, and since ELMo outperforms the other approaches for all important NLP tasks, we included the pre-trained ELMo contextual word representations into our benchmark to find out the embedding model for the retrieval task that produces the best result. The pre-trained ELMO model used the raw 1 Billion Word Benchmark dataset Chelba et al. (2013), and the vocabulary of 79347 tokens.

By having all the requirements set, it was time to compute ELMo representations for the SQuAD dev set using the pre-trained ELMo model. Before explaining the implementation strategies, we would like to mention the details of ELMo representation calculations. As they showed in their study, a $L$-layer bILM computes a set of $2L + 1$ representations for each token $t_i$.

$$\mathbf{L}_i = \{\boldsymbol{x}_i^{LM}, \overrightarrow{\boldsymbol{h}}_{i,j}^{LM}, \overleftarrow{\boldsymbol{h}}_{i,j}^{LM} | j = 1, ..., L\} = \{\overleftrightarrow{\boldsymbol{h}}_{i,j}^{LM} | j = 0, ..., L\}$$

where $\boldsymbol{x}_i^{LM}$, a context-indepedent token representation, is computed via a CNN over characters. Therefore $\boldsymbol{h}_{k,0}^{LM}$ is representing the token layer. These representations are then transmitted to $L$ layers of forward and backward LSTMs (or BiDirectional LSTMs-*biLSTMs*). Each LSTM layer provides a context-dependent vector representation $\overleftrightarrow{\boldsymbol{h}}_{i,j}^{LM}$ where $j = 1, ..., L$. While the top layer LSTM results, $\overrightarrow{\boldsymbol{h}}_{i,j}^{LM}$, presents the prediction of the next token $t_{i+1}$, likewise, $\overleftarrow{\boldsymbol{h}}_{i,j}^{LM}$ represents the prediction of the previous token $t_{i-1}$ by using a Softmax layer. The general approach would be a weighting of all biLM layers to get ELMo representations of the corpus. For our task, we generated the representations for the corpus, not only considering the results from the weighting of all biLM layers (*ELMo*), but also other individual layers such as token layer, biLSTM layer 1 and biLSTM layer 2.

As a result, we build the tensors based on tokens instead of documents; but documents can also be sliced from these tensors. To this end, we created a mapping dictionary for keeping the document's index and length (number of tokens). In this way a token is represented as the tensor of a shape $[1, 3, 1, 1024]$. To improve the traceability, the tensors were reshaped by swapping axis to become the tensor of a shape $[1, 3, 1024]$, then we stacked all these computed tokens to end up with the tensor of a shape $[416909, 3, 1024]$ where $416909$ is the number of tokens in the corpus (in other words, $416909$ represents the $12637$ documents in the corpus). Therefore, any document could be extracted from this tensor using the previously-created mapping dictionary. For example, let us assume that $doc_1$ has an index of $126$ and a length of $236$, so $doc_1$ can be extracted by slicing the tensor using such a format $[126 : 126 + 236, :, :]$. In terms of math expression, we ended up building a tensor $\mathsf{T}_{i,j,d}$ where $i$ is representing the total documents in the corpus, $j$ is representing the layers, and $d$ is a dimension space of a vector. $\mathsf{T}$ is based on tokens instead of documents, but documents can also be sliced from that tensor.

To achieve a good embedding, the main question was about identifying the most critical components that contain the most valuable information for the document. Since the new tensor of a shape $[12637, 3, 1024]$ has 3 components, in order to extract the most valuable slice(s), we defined a new weight tensor $\mathsf{W}_{1,j,1}$ where the $j$ had a same dimension as in the tensor $\mathsf{T}$. For our experiment, we set the $j$ to 3 as the same dimension value in which the pre-trained ELMo word representations were used. The elements of the vector $j$ of the tensor $\mathsf{W}$ was symbolized as $a$, $b$, and $c$, where $a + b + c = 1$. In order to find the right combination within the elements of the weight tensor $\mathsf{W}$ for the best embedding matrix $M$, we calculated the following function:

$$M' = ||mean(\mathsf{T} * \mathsf{W}; \boldsymbol{\theta} = 1)||$$

where $M' = \{M'_1, M'_2, \ldots, M'_n\}$, $\mathsf{W} = \{\mathsf{W}'_1, \mathsf{W}'_2, \ldots, \mathsf{W}'_n\}$, $n$ is the number of documents in the set and $\boldsymbol{\theta}$ represents the axis argument of the function. Last, but not least, all candidate embedding was normalized by L2-Norm. With this setup, we were able to create the candidate embedding matrices represented by $M'$ for further experiments to get the ultimate best embedding $E$. Each embedding in the embedding matrix is symbolized by $f(x) \in \mathbb{R}^d$. Therefore, it was able to embed a document $x$ into a d-dimensional Euclidean space.

In order to capture the best combination of $\mathsf{W}'$ (in other words, finding the best $a$, $b$, and $c$ values), we calculated the pair-wise distances between questions and paragraphs embedding represented $Q'$ and $P'$ respectively that were sliced from $M'$. The performance of finding the correct question-paragraph embedding pair is measured by $recall@k$. The reason why we used the $recall@k$ is that we wanted to compute the hitting performance of the true paragraph (answers) to the particular question whether it is in the top-k closest paragraphs or not. We used the ranks of $1, 2, 3, 5, 10$, and, $50$. In order to calculate the recall values, we had to deal with $20M$ question-paragraph embedding pairs ( $10K$ questions x $2K$ answers) in the dev dataset. We primarily aimed to look at the Receiver Operating Characteristic (ROC) curve and using Area Under Curve (AUC) as the target metric for tuning the embedding. Therefore, we sorted the retrieved samples by their distance (the paragraphs to the question) and then we computed recall and precision at each threshold. After having the precision numbers from the computations, we observed that our precision numbers became very low values because of having only $10k$ correct pairs and a high amount of negative paragraphs. We decided to switch to other methods called Precision-Recall curve and Average Precision.

Additionally, we took one step further. We wanted to inject the inverse document frequency (IDF) weights of each token into the embedding $M$ to observe whether it creates any positive impact on the representation or not. Since the IDF is calculated using the following function: $IDF(t) = log_e(\frac{\#of documents}{\#of docs.w/term})$, we calculated the IDF weights from each tokenized documents.Then we multiplied newly-computed token IDF weights with the original token embedding extracted from the pre-trained model to have the final injected token embedding.

Similar to both the ELMo and extension of ELMo with IDF weights steps, we followed the same pipeline for the GloVe since it is one of the very important baselines for this type of research. We specially utilized the *"glove-840B-300d"* pre-trained word vectors where it was trained on using the common crawl within 840B tokens, 2.2M vocab, cased, 300d vectors. We created the GloVe representation of our corpus and also extended it with IDF weights.

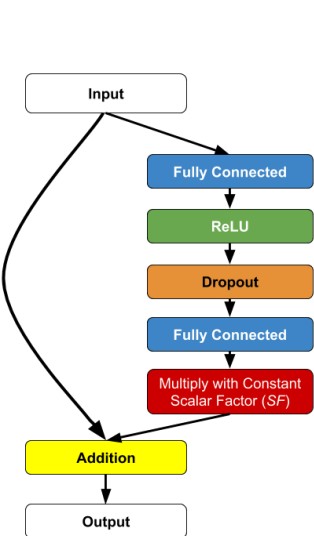

Figure 2: The layer architectures in the *Fully Connected Residual Retrieval Network* (**FCRR**).

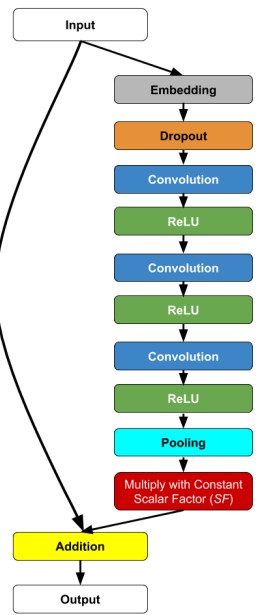

Figure 3: The layer architectures in the *Convolutional Residual Retrieval Network* (**ConvRR**).

## 3.2 RETRIEVAL MODEL

The retrieval model aims to improve the *recall@1* score by selecting the correct pair among all possible options. While pursuing this goal, it is developing embedding accuracy by minimizing the distance from the questions to the true paragraphs after embedding both questions and paragraphs to the semantic space.

The model architectures we designed to implement our idea are presented in Figure 2 and Figure 3. Question embedding $\boldsymbol{Q}^b$ *(b is representing the batch of question embedding)* were provided as inputs to the models. The output layer was wrapped with He et al. (2016)'s residual layer by scaling the previous layer's output with a constant scalar factor $sf$. Finally, the ultimate output from the residual layer was also normalized by L2. The normalized question embedding and the corresponding paragraph embedding are processed by the model that is optimized using a loss function $\mathscr{L}$. For this reason, we proposed and also utilized several loss functions. The main idea of all of this was to move the true $\boldsymbol{q} \in \boldsymbol{Q}^b, \boldsymbol{p} \in \boldsymbol{P}^b$ pairs closer to each other while keeping the wrong pairs further from each other. Using a constant margin value $m$ would be also helpful to create an enhanced effect on this goal.

The first function we defined was a **quadratic regression conditional loss**. With this $\mathscr{L}_{\text{quadratic regression conditionally}}$, we aimed to use a conditional margin.

$$m'(m, d) = \begin{cases} m, & \text{if } d > m. \\ 0, & \text{otherwise.} \end{cases} \quad (1)$$

$$\mathscr{L}_{\text{quadratic regression conditional}} = [||\boldsymbol{q} - \boldsymbol{p}||^2 - m]^+ + m'(m, ||\boldsymbol{q} - \boldsymbol{p}||^2)$$

The latter loss function we utilized was the **triplet loss**. With this setup, we not only consider the positive pairs but also negative pairs in the batch. Our aim was that a particular question $\boldsymbol{q}_{\text{anchor}}$ would be a question close in proximity to a paragraph $\boldsymbol{p}_{\text{positive}}$ as the answer to the same question than to any paragraph $\boldsymbol{p}_{\text{negative}}$ as they are answers to other questions. The key point of the $\mathscr{L}_{\text{triplet}}$ was building the correct triplet structure which should not meet the restraint of the following equation easily $||\boldsymbol{q}_{\text{anchor}} - \boldsymbol{p}_{\text{positive}}||^2 + m < ||\boldsymbol{q}_{\text{anchor}} - \boldsymbol{p}_{\text{negative}}||^2$. Therefore, for each anchor, we took the

positive $p_{\text{positive}}$ in such a way $\arg\max_{p_{\text{positive}}} ||q_{\text{anchor}} - p_{\text{positive}}||^2$ and likewise the hardest negative $p_{\text{negative}}$ in such a way that $\arg\min_{p_{\text{negative}}} ||q_{\text{anchor}} - p_{\text{negative}}||^2$ to form a triplet. This triplet selection strategy is called *hard triplets mining*.

$$\mathcal{L}_{\text{triplet}} = [||q_{\text{anchor}} - p_{\text{positive}}||^2 - ||q_{\text{anchor}} - p_{\text{negative}}||^2 + m]^+$$

During all our experiments, we have utilized Kingma & Ba (2014)'s ADAM optimizer with a learning rate of $10^{-3}$. For the sake of equal comparison, we have fixed the seed of randomization. We have also observed that a weight decay $10^{-3}$ and/or a dropout $10^{-1}$ is the optimum value for all types of model architectures to tackle over-fitting. In addition to that, the batch size $b$ of question-paragraph embedding pairs has been set to $512$ during the training. We trained the networks with a range of between $50$ to $200$ iterations.

As specifically, we designed our FCRR model as a stack of multi-layers. During the observation of the experiments, the simple but the efficient configuration was emerged as 2-layers. Likewise, we used $1024$ filters, $5$ as a length of the 1D convolution window, and $2$ as a length of the convolution in our ConvRR model. Last, but not least, the scaling factor is set to $1$ for both models.

## 4 EXPERIMENT

We did an empirical study of the proposed question-answering method in comparison with major state-of-the-art methods in terms of both text embedding and question answering. In the subsequent sections, we are going to describe details for the dataset, the model comparison and the results.

### 4.1 SQuAD DATASET

For the empirical study of the performance of text embedding and question answering, we use the Stanford Question Answering Dataset (SQuAD) Rajpurkar et al. (2016). The SQuAD is a benchmark dataset, consisting of questions posed by crowd-workers on a set of Wikipedia articles, where the answer to every question is a segment of text from the corresponding reading passage.

Table 1: Number of documents in the SQuAD Dataset.

| Set Name | # of Ques. | # of Par. | # of Tot. Cont |
|----------|-----------|-----------|----------------|
| Dev | 10570 | 2067 | 12637 |
| Train | 87599 | 18896 | 106495 |

### 4.2 EMBEDDING COMPARISON

We wanted to observe trade-off between Precision and Recall and AP value to compare performances of each candidate embedding $M'$. $recall@k$ was calculated for each embedding matrix $M'$ by using grid search over the $a, b, c$ components. See Table 2.

Table 2: The grid search results for both $recall@1$ and average $recalls$ among top 1, 2, 5, 10, 20, 50 recalls to find the best layer for the task.

| Layer Name | $W'$ Configuration | $recall@1$ | $avg\ recalls$ |
|------------|--------------------|------------|----------------|
| Token Layer | $a = 1, b = 0, c = 0$ | $41.50\,\% \pm 0.03\,\%$ | $66.40\,\% \pm 0.02\,\%$ |
| biLSTM Layer 1 | $a = 0, b = 1, c = 0$ | $26.20\,\% \pm 0.01\,\%$ | $54.30\,\% \pm 0.05\,\%$ |
| biLSTM Layer 2 | $a = 0, b = 0, c = 1$ | $19.60\,\% \pm 0.07\,\%$ | $48.00\,\% \pm 0.04\,\%$ |
| ELMO | $a = 0.33, b = 0.33, c = 0.33$ | $21.20\,\% \pm 0.01\,\%$ | $48.30\,\% \pm 0.02\,\%$ |

The best component combination emerged as $a = 1$, $b = 0$, $c = 0$ that provides the best setting to represent the corpus. In other words, token layer is the most useful layer. Finally, question and paragraph embedding are represented $Q = \{q_1, q_2, \ldots, q_x\}$ and $P = \{p_1, p_2, \ldots, p_{(n-x)}\}$,

where $x$ represents the number of questions within the corpus. In other terms $[0 : 10570, 1024]$ belongs to questions while $[10570 :, 1024]$ belongs to paragraphs for the dev dataset.

We can now start to compare the results of the $recall@k$ of the pair-wise embedding that are derived from different models for the **dev corpus** in order to measure the embedding quality. The $recall@k$ tables and the Precision-Recall curve from each of the embedding structure are presented in the section below.

Table 3: $recall@k$s of GloVe.

| $recall@k$ | # of docs | % |
|---|---|---|
| 1 | 3246 | 30.7 % |
| 2 | 4318 | 40.8 % |
| 5 | 5713 | 54.0 % |
| 10 | 6747 | 63.8 % |
| 20 | 7602 | 71.9 % |
| 50 | 8524 | 80.6 % |

Figure 4: P-R Curve for GloVe.

Table 4: $recall@k$s of Extension of GloVe with IDF weights.

| $recall@k$ | # of docs | % |
|---|---|---|
| 1 | 3688 | 34.8 % |
| 2 | 4819 | 45.5 % |
| 5 | 6293 | 59.5 % |
| 10 | 7251 | 68.5 % |
| 20 | 8076 | 76.4 % |
| 50 | 8919 | 84.3 % |

Figure 5: P-R Curve for extension of GloVe with IDF weights.

Table 5: $recall@k$s of ELMo.

| $recall@k$ | # of docs | % |
|---|---|---|
| 1 | 4391 | 41.5 % |
| 2 | 5508 | 52.1 % |
| 5 | 6793 | 64.2 % |
| 10 | 7684 | 72.6 % |
| 20 | 8474 | 80.1 % |
| 50 | 9271 | 87.7 % |

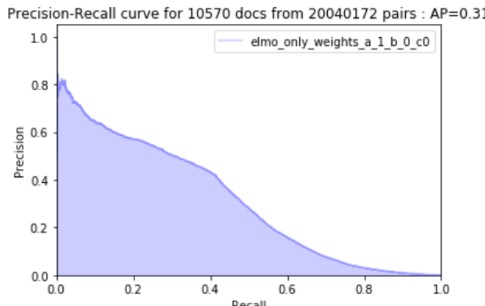

Figure 6: P-R Curve for ELMo.

Table 6: $recall@k$s of Extension of ELMo with IDF weights.

| $recall@k$ | # of docs | % |
|---|---|---|
| 1 | 4755 | 44.9 % |
| 2 | 5933 | 56.3 % |
| 5 | 7208 | 68.1 % |
| 10 | 8040 | 76.0 % |
| 20 | 8806 | 83.3 % |
| 50 | 9492 | 89.8 % |

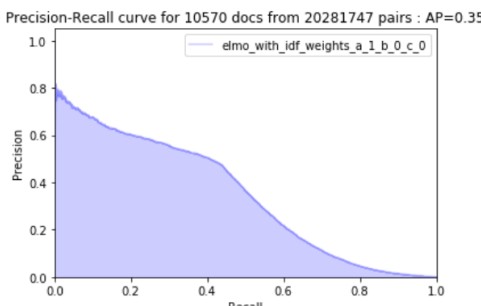

Figure 7: P-R Curve for extension of ELMo with IDF weights.

The average-precision scores of each embedding structure is listed in Table 7.

Table 7: Average Precision List.

| Model Name | Average Precision (AP) |
|---|---|
| GloVe | 0.17 |
| Extension GloVe W/ IDF w. | 0.22 |
| ELMO | 0.31 |
| Extension ELMO W/ IDF w. | 0.35 |

By following the same pipeline, we applied the defined best model to the **train set**. As an initial step, we randomly selected 5K question embedding and corresponding paragraph embedding from the corpus as our validation set. The remaining part of the set is defined to be used as a training data. The results from the best model (*Extension of ELMo w/ IDF weights*) and baseline model (*Only ELMo*) are presented in Table 8.

Table 8: $recall@k$s of Extension of ELMo w/ IDF weights and Only ELMo from the train set..

| $recall@k$ | ELMo | | Ext. of ELMo w/ IDF | |
|---|---|---|---|---|
| | # of docs | % | # of docs | % |
| 1 | 1445 | 30.38 % | 1723 | 34.46 % |
| 2 | 1918 | 38.36 % | 2128 | 42.56 % |
| 5 | 2424 | 48.48 % | 2688 | 53.76 % |
| 10 | 2820 | 56.40 % | 3045 | 60.90 % |
| 20 | 3143 | 62.86 % | 3388 | 67.76 % |
| 50 | 3545 | 70.90 % | 3792 | 75.84 % |

As presented in Table 8, we were able to improve the representation performance of the embeddings increased by $4\%$ on a $recall@1$ and $4.7\%$ on an *average recall* for our task by injecting the IDF weights into the ELMo token embedding.

### 4.3 MODEL FOR RETRIEVAL

After having the best possible results as a new baseline, we aimed to improve the representation quality of the question embedding. In order to train our models for the retrieval task, we decided to **merge the SQuAD dev and train datasets** to create the larger corpus. By executing that step, we have a total of 98169 questions and 20963 paragraphs. We randomly selected 5K question embedding and corresponding paragraph embedding from the corpus as our validation set for the recall task and similarly 10K as our validation set for the loss calculation task. The remaining part of the set is defined as the training data.

The intuition was that the correct pairs should be closer in proximity to one another so that closest paragraph embedding to the question embedding would be correct in an ideal scenario. With this goal; *1)* the question embedding is fine-tuned by our retrieval models so that the closest paragraph embedding would be the true pair for the corresponding question embedding, *2)* similarly, the paragraph embedding is also fine-tuned by our retrieval model so that the closest question embedding would be the true pair for the corresponding paragraph embedding, and, finally *3)* on top of previous steps, we further fine-tuned question embedding using the combination of improved question and paragraph embedding stated at step 1 & 2.

Table 9: The highest $recall@1$ and average $recalls$ among top 1, 2, 5, 10, 20, 50 results from the best models including the baseline model for question embedding.

| Model | Question Embedding | | | |
|---|---|---|---|---|
| | $recall@1$ | improv. (%) | avg $recalls$ | improv. (%) |
| Only ELMo | $28.40\% \pm 0.06\%$ | - | $49.70\% \pm 0.02\%$ | - |
| ELMo w/ IDF | $32.60\% \pm 0.04\%$ | $4.2\% \pm 0.1\%$ | $54.20\% \pm 0.06\%$ | $4.50\% \pm 0.08\%$ |
| FCRR | $40.50\% \pm 0.02\%$ | $12.10\% \pm 0.08\%$ | $64.30\% \pm 0.03\%$ | $14.60\% \pm 0.05\%$ |
| FCRR + ConvRR | $41.60\% \pm 0.06\%$ | $13.20\% \pm 0.12\%$ | $65.50\% \pm 0.04\%$ | $15.80\% \pm 0.06\%$ |
| FCRR + ConvRR + w/ fine-tuned Paragraph embed. | $41.96\% \pm 0.07\%$ | $13.56\% \pm 0.15\%$ | $65.80\% \pm 0.06\%$ | $16.10\% \pm 0.08\%$ |

Table 10: The highest $recall@1$ and average $recalls$ among top 1, 2, 5, 10, 20, 50 results from the best models including the baseline model for paragraph embedding.

| Model | Paragraph Embedding | | | |
|---|---|---|---|---|
| | $recall@1$ | improv. (%) | avg $recalls$ | improv. (%) |
| Only ELMo | $6.70\% \pm 0.07\%$ | - | $22.50\% \pm 0.09\%$ | - |
| ELMo w/ IDF | $8.80\% \pm 0.02\%$ | $2.10\% \pm 0.09\%$ | $26.70\% \pm 0.04\%$ | $4.20\% \pm 0.13\%$ |
| FCRR | $13.60\% \pm 0.05\%$ | $6.90\% \pm 0.12\%$ | $41.40\% \pm 0.08\%$ | $18.90\% \pm 0.17\%$ |
| FCRR + ConvRR | $14.00\% \pm 0.02\%$ | $7.30\% \pm 0.09\%$ | $42.10\% \pm 0.04\%$ | $19.60\% \pm 0.13\%$ |
| FCRR + ConvRR + w/ fine-tuned Question embed. | $14.65\% \pm 0.01\%$ | $7.95\% \pm 0.08\%$ | $42.70\% \pm 0.03\%$ | $20.20\% \pm 0.12\%$ |

As shown in Table 9 and Table 10, we were able to improve the question and paragraph embedding such a way that the retrieval performance for the question side increased by $9.36\% \pm 0.11\%$ on a *recall@1* result and $11.60\% \pm 0.12\%$ on an *average recall* compared to our new baseline that is an extension of ELMo w/ IDF. In addition to that, if we compare our best results with the original ELMo baseline results, we can observe that we also improved the representation performance increased by $13.56\% \pm 0.15\%$ on a *recall@1* and $16.10\% \pm 0.08\%$ on an *average recall*. Similarly, the retrieval performance for the paragraph side increased by $8.85\% \pm 0.03\%$ on a *recall@1* result and $16.00\% \pm 0.07\%$ on an *average recall* compared to our new baseline that is an extension of ELMo w/ IDF. In addition to that, if we compare our best results with the original ELMo baseline results, we can observe that we also improved the representation performance increased by $7.95\% \pm 0.08\%$ on a *recall@1* and $20.20\% \pm 0.12\%$ on an *average recall*

On the other hand, $\mathscr{L}_{\text{quadratic regression conditional}}$ didn't show any improved performance compared to $\mathscr{L}_{\text{triplet}}$. Last, but not least, during our experiments, we also wanted to compare the quality of our embedding with the embedding derived from USE. Although our best model with the $\mathscr{L}_{\text{triplet}}$ was able to improve the representation performance of the USE document embedding for the retrieval task, it was still far from achieving our state-of-the-art fine-tuned result.

## 5 CONCLUSION

We developed a new question answering framework for retrieval answers from a knowledge base. The critical part of the framework is text embedding. The state-of-the-art embedding uses the output of the embedding model as the representation. We developed a representation selection method for determining the optimal combination of representations from a multi-layered language model. In addition, we designed deep learning based retrieval models, which further improved the performance of question answering by optimizing the distance from the source to the target. The empirical study using the SQuAD benchmark dataset shows a significant performance gain in terms of the recall. In the future, we plan to apply the proposed framework for other information retrieval and ranking tasks. We also want to improve the performance of the current retrieval models by applying and developing new loss functions.

ACKNOWLEDGMENTS

This study used the Google Cloud Computing Platform (GCP) which is supported by Google AI research grant.

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
