# OpenReview forum: "Text Embeddings for Retrieval from a Large Knowledge Base"
_ICLR.cc/2019/Conference_

### Official Review · AnonReviewer1 · 2018-11-02
**Interesting results but need more experiments and clearer explanation**

**Rating:** 5
**Confidence:** 4

**Review:**

This paper proposed a retrieval model based on the residual network and evaluated the use of ELMo word embedding with/without IDF weight. The results showed that there are significant gain when adding the residual network on top of the word embedding.

Pros:
* This work set a strong baseline for the retrieving target paragraph for question answering on the SQuAD dataset.
* The experiments were sounds and leverage interesting points -- the use of word embedding itself as the feature representation didn't have as much impact to retrieval performance as the distance function.
* The studied problem -- retrieval for answering question rather than getting the most relevant document worth more attention.

Cons:
* The motivation of using the word embedding and contextual word embedding over the TF-IDF feature wasn't clear. Results on using simple feature like TF-IDF maybe useful to give readers better judgement of the use of word embedding.
* The choice of dataset, SQuAD over more retrieval based QA like TrivialQA also wasn't strongly motivated. Also, it would be nice to see how the QA result would be improve with better retrieval model.
* Another use of TF-IDF/IDF and embedding is to use TF-IDF/IDF to identify the related document and then use word embedding to resolve semantic ambiguity. Do you have theoretical/empirical reason why this shouldn’t be considered?

Comment on writing:
    - In Section 3.1: the dimension of the tensor should reflect the meaning (vocab size, embedding size or the number of documents) rather than numbers.
    - In Section 3.1: since the weighting for each document is not shared, it would be clearer to just use M and W for each document instead of M’, W'
    - In Section 3.1: Evaluation metrics, e.g., recall@k, ROC, AUC; technical details, for example, tensor dimension, optimizer hyperparameters should be moved to the experiment section

---

### Official Review · AnonReviewer3 · 2018-11-02
**Not ready for publication**

**Rating:** 3
**Confidence:** 5

**Review:**

This paper tries to study retrieval methods for multi-paragraph / multi-document reading comprehension.  The basic approach is to embed the question and the paragraph and train a system to put the correct paragraph close to the question.  I had a very hard time following the details of the proposed approach for this, however, and I still don't really understand what the authors are proposing.

This paper is not ready for publication.  The exposition is not at all clear and needs substantial rewriting.  Additionally, the evaluation done in the paper is not well-justified.  I do not know what "paragraph-side" means, but I assume that means you are trying to retrieve the question given the paragraph.  Why?  There were no standard baselines compared against, like a simple IR system (Lucene).  And I expected to see actual impact of the retrieved results on downstream QA performance of a system like Chen et al.'s, or Clark and Gardner 2018.  Even if you have a slightly better ranking of the retrieved paragraphs, it's not clear to me that this will improve performance, if the downstream method is properly calibrated to handle multiple paragraphs (see Clark and Gardner 2018).

A few writing suggestions for the authors, for next time:

This paper does not follow the typical flow of an academic paper.  It reads too much like a logbook of what you did, presented chronologically, instead of presenting the ideas in a coherent sequence.  Part of this is just simple wording fixes (e.g., avoid things like "it was time to compute ELMo representations" - this isn't a logbook).  Also, all of the shape comments and numerical details at the top of page 4 are out of place.  Describe your method first in general terms, then give experimental details (like corpus size, etc.) later.  I suggest reading the award-winning papers at various conferences to get a sense of how these papers are typically structured and phrased.

Section 2: A full page dedicated to the history of word embeddings is entirely unnecessary for this paper.  This is not a survey on word embeddings.  It's much more useful to the reader to give pointers to multiple connection points between your work and the rest of the literature.  You could have given a paragraph to the most relevant embedding techniques, a paragraph to the most relevant retrieval / multi-paragraph techniques (e.g., Clark and Gardner 2018, which is very relevant, along with Chen et al., TriviaQA, others), and a paragraph to distance metric learning.

---

### Official Review · AnonReviewer2 · 2018-11-03
**Good motivation, weak organization, unclear results**

**Rating:** 3
**Confidence:** 4

**Review:**

Summary:
This paper proposes to reformulate the QA task in SQUAD as a retrieval task, i.e., using question as query and paragraphs as candidate results to be ranked.  Authors makes some modifications to elmo model to create better word embedding for the ranking task. Authors have mentioned and are aware of open domain QA methodologies (e.g., DrQA).

Pros:
- The general idea is interesting, to reformulate any QA task as a ranking task

Cons:
- The methodology and task are not clear. Authors have reformulated QA in SQUAD as as ranking and never compared the results of the proposed model with other QA systems. If authors want to solve a pure ranking problem why they do not compare their methods with other ranking methods/datasets.
- The novelty: The novelty is not significant. Although modifications to ELMO are interesting.
- Results: Why authors have not compared their work with DrQA?

---

### Meta-Review · Area_Chair1 · 2018-12-13
**Serious evaluation issues**

**Confidence:** 4
**Recommendation:** Reject

**Metareview:**

I have to agree with the reviewers here and unfortunately recommend a rejection.

The methodology and task are not clear. Authors have reformulated QA in SQUAD as as ranking and never compared the results of the proposed model with other QA systems. If authors want to solve a pure ranking problem why they do not compare their methods with other ranking methods/datasets.